# Effect of Laser Micro-Texturing on Laser Joining of Carbon Fiber Reinforced Thermosetting Composites to TC4 Alloy

**DOI:** 10.3390/ma16010270

**Published:** 2022-12-27

**Authors:** Junke Jiao, Jihao Xu, Chenghu Jing, Xiangyu Cheng, Di Wu, Haolei Ru, Kun Zeng, Liyuan Sheng

**Affiliations:** 1School of Mechanical Engineering, Yangzhou University, Yangzhou 225009, China; 2PKU-HKUST Shenzhen-Hong Kong Institution, Shenzhen 518057, China; 3Ningbo Institute of Materials Technology and Engineering, CAS, Ningbo 315201, China; 4Yangzhou Hanjiang Yangzi Automobile Interior Decoration Co., Ltd., Yangzhou 225009, China

**Keywords:** CFTRS, TC4 alloy, laser welding, laser micro-texturing, temperature field, finite element mode

## Abstract

Carbon fiber reinforced thermosetting composites (CFRTS) and TC4 alloy are important structural materials for lightweight manufacturing. The hybrid structure of these two materials has been widely used in the aerospace field. However, the CFRTS-TC4 alloy joint formed by the traditional connection method has many challenges, such as poor environmental adaptability and stress concentration. Laser micro-texturing of metal surface-assisted laser connection of CFRTS and TC4 alloy has great potential in improving joint strength. In order to study the effect of laser micro-texturing on the laser bonding of CFRTS and TC4 alloy, the simulation and experimental research of laser welding of TC4 alloy and CFRTS based on laser micro-textures with different scanning spacings were carried out, and the interface hybrid pretreatment method of laser cleaning and laser plastic-covered treatment was introduced to assist the high-quality laser bonding of heterogeneous joints. The results showed that the established finite element model of CFRTS-TC4 alloy laser welding can predict the temperature field distribution of the joint during the welding process and reflect the forming mechanism of the joint. The laser micro-textures with different scanning spacings will lead to a difference in the temperature field distribution on the polyamide (PA6) interface, which leads to a change in heat input on the CFRTS surface. When the laser scanning spacing is 0.3 mm, the joint strength can reach 14.3 MPa. The failure mechanism of the joint mainly includes the cohesive failure of the internal tear of the carbon fiber and the interfacial failure of the interface between the PA6 resin and the TC4 alloy.

## 1. Introduction

Carbon fiber reinforced thermosetting composites (CFRTS) are widely used in the aerospace field due to their low density, high specific strength, fatigue resistance, and corrosion resistance. As a lightweight metal material, TC4 alloy has excellent mechanical properties and has become an indispensable structural material in the aerospace field. In recent years, the aviation industry has been using CFRTS and TC4 alloy heterogeneous composite structures, such as aircraft fuselage and wing panels, landing gear and hood, and other structural parts, to effectively reduce the overall weight of the aircraft, energy consumption, and cost [1,2,3]. With the rapid expansion of the aerospace field, there is a higher standard for the performance of heterogeneous hybrid structures of CFRTS and TC4 alloy. However, due to the large differences in the physical and chemical properties of the two materials, the traditional connection methods, such as mechanical connection and bonding, have many problems, such as stress concentration and poor environmental adaptability, which cannot meet the current requirements [4]. Therefore, exploring and refining the connection process between CFRTS and TC4 alloy to establish a high-quality connection between these two materials has become key to improving the performance of the heterostructure.

Laser welding technology is a non-contact, high-energy, and efficient connection technology. It has the characteristics of fast speed, small deformation, and easy automation. It has great potential in the high-quality connection of composite materials and metals [5]. It uses a high-energy density laser beam to irradiate the metal surface, and the metal absorbs the laser energy and transfers the heat generated to the bonding interface to heat the resin matrix of the molten composite material. The molten resin combines with the metal under the action of molecular force and chemical bonding to realize a connection between the two materials [6]. Lambiase [7] carried out laser welding experiments on AISI304 stainless steel and polycarbonate sheet (PC). It was found that at high laser power and low welding speed, excessive heat input leads to a large number of bubble defects at the bonding interface. At low laser power and high welding speed, insufficient heat input leads to insufficient bonding between PC and stainless steel, resulting in an interface gap and a negative impact on joint strength. In the laser welding experiment of carbon fiber reinforced thermoplastic composite (CFRTP) and aluminum alloy, Li [8] further proved that the bubble defects generated during the welding process lead to some unfused areas between CFRTP and aluminum alloy, reducing the mechanical properties of the joint. In the laser welding experiment of CFRTP and TC4 alloy, Tao [9] analyzed the fracture morphology of the joint and found that the fracture form of the joint was a mixed fracture feature of cohesive failure and adhesion failure. Jiao [10,11] studied the influence of welding process parameters, such as laser power, welding speed, and fixture pressure on the size of the thermal defect zone and connection strength of CFRTP and stainless steel laser connection joints. The results showed that laser power and welding speed have the most significant influence on the joints. By optimizing the process parameters and adding a resin layer between the joint surfaces, the welding defects can be significantly reduced.

To improve the strength of laser bonding between composite materials and metals, researchers have proposed a series of metal surface pretreatment methods, such as mechanical treatment, chemical treatment, laser micro-texturing, etc. Among them, mechanical treatment and chemical treatment are not suitable for regulation. The randomness of the treated metal surface is large, and the improvement in the joint strength is limited. Laser micro-texturing can prepare specific micro-textures of micron to millimeter magnitude on the metal surface, and controlling the laser process parameters allows for a wide range of control over the shape and distribution of the micro-textures on the metal surface [12,13], resulting in higher joint strength [14]. Zhang [15] used a fiber laser to prepare protrusion structures on the metal surface. By adjusting the scanning times and scanning spacing of the laser beam to control the structural height and structural density, the effect of the protrusion structure on the laser bonding strength between A7050 aluminum alloy and CFRTP was studied. It was found that the joint strength increases almost linearly with an increase in the density of the protrusion structure. When the structural density was 1.11 mm^−2^, the joint strength was as high as 39.0 MPa, which was about 4.5 times that of the untreated joint. Heckert [16] used different types of laser to prepare different scales of micro-textures on the surface of aluminum alloy and carried out laser joining of aluminum alloy and glass fiber composites. The results showed that the joint strength of aluminum alloy and glass fiber composites formed by nanosecond laser micro-textured aluminum alloy was up to 42 MPa. In the laser welding of CFRTP and TC4 alloy, Tan [17] prepared laser micro-textures on the surface of TC4 alloy using a nanosecond laser system and studied the effect of micro-texture depth on the strength of the joint. The results showed that the shear force of the joint increases by about 156% after laser micro-texturing, and with the increase in the micro-texture depth, the strength of the joint increases first and then decreases. When the micro-texture depth is 100 μm, the maximum shear force can reach 2621 N. Liu [18,19] found that laser micro-textures significantly improve the roughness and wettability of the TC4 surface and increase the bonding area of the interface, thus enhancing the shear force of the joint. At the same time, it was also found that the groove width of micro-textures has a significant effect on the joint strength. When the groove width was 0.3 mm, the maximum shear force of the joint could reach 2698.8 N, which was 2.1 times that of the untreated joint.

The temperature of the joint is the key to the welding quality in the process of laser welding. Mastering the temperature field distribution of the joint is helpful not only to study the thermal effect in the welding process and the forming mechanism of the joint, but also to optimize the welding parameters and provide theoretical support for obtaining high-quality joints. However, it is very difficult to directly detect the temperature distribution inside a joint, and numerical simulation technology is an effective means of obtaining the temperature field distribution of a joint. In recent years, researchers have conducted exploratory research on the numerical simulation technology of the temperature field of laser welding of composite materials and metals. Lambiase [20,21] studied the temperature field of laser welding of polycarbonate and AISI304 stainless steel by combining experimental measurement with finite element simulation and obtained good correspondence, which proved the feasibility of simulation. In the follow-up study, the optimization of welding process parameters was realized based on the simulation of the temperature field. Tan [22] established a finite element model of laser welding of CFRTP and TC4 alloy and studied the influence of laser scanning speed on the connection process of CFRTP and TC4 alloy. The results show that with an increase in scanning speed, the melting range of CFRTP increases first and then decreases, which further explains the fluctuation of the shear resistance of CFRTP-TC4 joints. Liu [23] established the thermal contact model of laser transmission welding to solve the problem of low accuracy of the finite element model of traditional laser transmission welding and predicted the weld temperature profile. The results showed that the thermal contact model was more consistent with the real model. Jiao [24] established a thermal contact model of CFRTP and stainless steel laser welding considering the thermal contact resistance between joint surfaces in the actual welding process, which reduced the relative error of the traditional finite element model to 6.7%. Based on the thermal contact model, the effects of laser power, welding speed, and clamping pressure on the temperature field of the joint were studied, which provided theoretical support for the subsequent experiments.

In summary, laser micro-textured metal surface can significantly improve the bonding strength of heterogeneous joints and contribute to the acquisition of high-quality welded joints. Using numerical simulation technology can predict the joint temperature field and provide good theoretical support for experimental research. In the traditional numerical model of the temperature field, the influence of welding process parameters, including laser power, welding speed, and fixture pressure, on the temperature field is generally considered. However, in addition to the above factors, the laser micro-textures of the metal surface may also affect the temperature field of the joint. This is mainly because the micro-textures change the contact area between the metal surface and the molten resin, causing the heat change per unit area. At present, no research has shown whether laser micro-textures affect the temperature field of the joint during laser welding. Based on this, a finite element model of the laser welding temperature field of TC4 alloy and CFRTS based on laser micro-textures with different scanning spacings is established in this paper. The finite element method was used to calculate and analyze the temperature field of the joint. The influence of laser micro-textures with different scanning spacings on the temperature field of the joint is explored, and the thermal effect in the welding process and the forming mechanism of the joint are explained. In addition, laser welding experiments of CFRTS and TC4 alloy were carried out to verify the reliability of the simulation results, and the effects of laser micro-textures with different scanning spacings on the mechanical properties of joints were further explored. The related work will lay a theoretical and technical foundation for high-quality laser welding of CFRTS and TC4 alloy.

## 2. Materials and Methods

### 2.1. Materials

The experimental materials were a TC4 alloy plate and a CFRTS plate with a size of 50 mm × 25 mm × 2 mm. The CFRTS was formed by a hot pressing process with epoxy resin as matrix and T300 unidirectional carbon fiber as reinforcing material, and the content of epoxy resin was about 38%. The thermoplastic polyamide resin (PA6) powder was selected as the plastic-covered layer with a thickness of about 100 μm. Table 1 gives the thermal physical parameters of epoxy resin, T300, and PA6 [25]. Table 2 gives the thermal physical parameters of TC4 alloy at different temperatures [22].

### 2.2. Experimental Procedure

Since the epoxy resin matrix of CFRTS is thermosetting resin, it cannot be remelted and cannot be directly connected with TC4 alloy. Therefore, in order to realize the high-quality laser welding of CFRTS and TC4 alloy, the interface hybrid pretreatment process of laser cleaning and laser plastic-covered treatment is adopted to pretreat the surface of CFRTS and TC4 alloy before welding, as shown in Figure 1.

The first step is to pretreat the surface of the TC4 alloy by a nanosecond laser processing system. As shown in Figure 2a, the system mainly includes a nanosecond laser, a control system, and a scanning galvanometer. The wavelength, pulse width, pulse frequency, and spot diameter of the nanosecond laser are 1064 nm, 100 ns, 20 kHz, and 0.1 mm, respectively. The laser beam is focused by the F-theta lens in the scanning galvanometer. The laser power can be adjusted in the range of 10–100%, and the maximum power is 100 W. Combined with our previous experimental studies [26], the laser power, scanning speed, and scanning times were set as 95 W, 2000 mm·s^−1^, and 5, respectively. The surface of the TC4 alloy was laser micro-textured by adjusting the scanning spacing *d* to form three groups of micro-textures: *d* = 0.1 mm, *d* = 0.3 mm, and *d* = 0.5 mm. Then, a layer of thermoplastic resin powder with a thickness of 100 μm was laid on the surface of the laser micro-textured TC4 alloy. The resin powder was scanned using a nanosecond laser beam to absorb laser energy, melt the powder, and embed it in the micro-textures. After cooling and solidification, a plastic-covered layer was formed on the surface of the TC4 alloy. Based on previous experimental research [26], the laser power, scanning speed, scanning spacing, and scanning times of the laser plastic-covered layer were selected as 35 W, 200 mm·s^−1^, 0.3 mm, and 5, respectively. Under these conditions, the surface roughness of the plastic-covered layer is low, the PA6 resin is fully melted, and no ablation occurs.

The second step was to use the picosecond laser processing system to perform laser cleaning pretreatment on the CFRTS surface. As shown in Figure 2b, the system mainly includes a picosecond laser, a scanning galvanometer, and a control system. The wavelength, pulse width, frequency, and spot diameter of the picosecond laser were 532 nm, 10 ps, 50 kHz, and 0.03 mm, respectively. The laser power could be adjusted in the range of 30–100%, and the maximum power was 12 W. The basic principle of laser cleaning is that when the laser beam acts on the surface of CFRTS, the laser energy is absorbed by CFRTS in a nonlinear form, and ionization and avalanche ionization occur, resulting in high-density and high-temperature plasma. Due to the high pressure inside the CFRTS, the plasma splashes from it, thereby removing the epoxy resin on the surface and fully exposing the internal carbon fiber. Based on previous experimental research [27], in order to remove the epoxy resin on the surface of CFRTS and ensure the integrity of the internal carbon fiber, the laser power, scanning speed, scanning spacing, and scanning times of laser cleaning were selected as 1.32 W, 2500 mm·s^−1^, 0.03 mm, and 9, respectively.

After the interface hybrid pretreatment process, the laser-assisted joining of CFRTS and TC4 alloy was carried out based on the same nanosecond laser processing system. Previous studies have found that a high-speed rotating laser beam can effectively reduce the thermal damage caused by laser heating on TC4 alloy and improve the bonding area of TC4 alloy and composite materials, thereby improving the strength of the joint [26]. Therefore, this paper uses high-speed rotating laser welding technology to join CFRTS and TC4 alloy. The basic principle is shown in Figure 1. The surface of the TC4 alloy absorbs energy under laser irradiation, and the heat generated is transferred to the bonding interface by heat conduction to melt the PA6 resin again. The molten resin flows fully into the laser cleaning area of CFRTS under the action of fixture pressure and is fully combined with the surface of the TC4 alloy. After cooling and solidification, the CFRTS-TC4 alloy welded joint is formed. Table 3 shows the process parameters of CFRTS and TC4 laser welding, which were optimized in previous work [26], where the laser power is *P*, scanning speed is *V*, scanning spacing is *D*, laser beam amplitude is *a*, spot radius is *r*, and fixture pressure is *P*_n_.

### 2.3. Analysis Methods

The morphology and size of micro-textures on the surface of the TC4 alloy were extracted using a laser confocal microscope. The forming quality of the joint and the morphology of the fracture surface were detected and analyzed using a scanning electron microscope. In order to study the effect of micro-textures with different scanning spacings on the mechanical properties of CFRTS-TC4 joints, the tensile test of the joints was carried out at room temperature using a universal material testing machine. Before the start of the experiment, the corresponding gaskets were installed at both ends of the sample to make the tensile sample, as shown in Figure 3. The tensile speed of the testing machine was set to 2 mm·min^−1^, and the preload was 0.5 MPa. Each group of samples was repeated three times to ensure the accuracy of the data. In this study, the prediction and analysis of the CFRTS-TC4 joint temperature field in the welding process, mainly through finite element simulation, is described in detail in the next section.

### 2.4. Finite Element Simulation

#### 2.4.1. Establishment of Heat Transfer Model

The moving path of the laser heat source is shown in Figure 4. It can be seen that the moving trajectory of the laser beam on the surface of the TC4 alloy includes horizontal displacement and periodic circular motion. Therefore, a curve similar to the spiral line is formed. The fitting formula of the curve can be expressed as follows.
(1)$$\{\begin{array}{l} x(t)=x_{0}+vt+a\cos(2\pi ft+\varphi_{0})\\ y(t)=y_{0}+a\sin(2\pi ft+\varphi_{0}) \end{array}$$
where *v* represents the forward moving speed of the laser beam relative to the *x*-axis, and its relationship with the laser beam rotation frequency *f* and the scanning interval *D* satisfies Formula (2). *φ*_0_ is the initial phase angle of the rotating laser beam, and this paper takes *φ*_0_ = π. *x*_0_ is the initial coordinate of the main direction of welding, and *y*_0_ is the initial coordinate perpendicular to the main direction of welding, where the initial coordinate is the origin of the coordinate.
(2)v=f×D

Since the absorption coefficient of TC4 alloy to laser beam is small, the laser heat source model is approximately treated as a Gaussian distributed surface heat source model [28]. The heat transfer mathematical model of CFRTS-TC4 alloy high-speed rotating laser welding can be expressed as:(3)$$\{\begin{array}{l} q(x,y,z,t)=\frac{(1-R)P}{\pi r^{2}}\exp\left\{-\frac{{\left[x+a\cos(2\pi ft)-vt\right]}^{2}+{\left[y+a\sin(2\pi ft)\right]}^{2}}{r^{2}}\right\}\\ T(x,y,z,0)=T_{0}\;\;\;\;\;\;\;\;\;\;\;\;\;\;\;\;\;\;\;\;\;\;\;\;\;\;\;\;\;\;\;\;\;\;\;\;\;t=0\\ -k\frac{\partial T}{\partial n}=q(x,y,z,t)\;\;\;\;\;\;\;\;\;\;\;\;\;\;\;\;\;\;\;\;\;\;\;\;\;\;\;\;\;\;\;\;z=0\\ -k\frac{\partial T}{\partial n}=h_{c}(T-T_{0})\;\;\;\;\;\;\;\;\;\;\;\;\;\;\;\;\;\;\;\mathrm{On}\;\mathrm{the}\;\mathrm{other}\;\mathrm{surfaces} \end{array}$$
where *q* is the heat flux, *P* is the laser output power, *r* is the effective radius of the laser beam, *k* is the thermal conductivity, *n* is the surface normal vector, and *R* is the reflectivity of the TC4 alloy to the laser. According to the actual situation, this paper assumes *R* = 55%, the initial welding temperature *T*_0_ = 25 °C, and the convective heat transfer coefficient *h*_c_ = 20 W·m^−2^·°C^−1^.

#### 2.4.2. Establishment of Finite Element Model

The geometric model of the CFRTS-TC4 joint was established by using the finite element software ANSYS19.0. As shown in Figure 3, the dimensions of TC4 alloy and CFRTS are both 50 mm × 25 mm × 2 mm, and the overlap area is 25 mm × 20 mm. The surface morphology of the TC4 alloy after laser micro-texturing is shown in Figure 5. In this paper, the micro-textures cell model is approximately established as a pyramid structure. The top edge length is *l*_1_, the bottom edge length is *l*_2_, the height is *h*, and the spacing between micro-textures is *w*. Table 4 gives the micro-texture sizes under different scanning spacings. The micro-texture geometric model with a scanning spacing of 0.3 mm is shown in Figure 6a. Figure 7a shows the surface morphology of the TC4 alloy after laser plastic-covered treatment. It can be seen that PA6 resin is embedded in the micro-textured groove and closely combined with TC4 alloy, thus establishing the anchorage model (Figure 6b) of the CFRTS-PA6 plastic-covered layer. Figure 7b shows the surface morphology of CFRTS after laser cleaning. The surface epoxy resin of CFRTS is removed in large quantities, and the internal carbon fiber is almost undamaged and has good integrity. Therefore, the influence of surface epoxy resin is not considered when modeling. The Solid70 tetrahedral thermal analysis unit is used to divide the model into free grids. In order to improve computational efficiency, only the lap area model is established. At the same time, in order to take into account the calculation accuracy, the grid of the connection interface is encrypted. The grid distribution is shown in Figure 6c. The loading of the moving heat source and the solution of the temperature field are realized by the APDL programming language. The heat source is calculated once every step of the movement, and cyclic loading and calculation are performed until the heat source completely leaves the TC4 alloy surface.

In this paper, three groups of finite element models were established according to the size of micro-textures under three groups of scanning spacings. In addition, a fourth group of finite element models without considering micro-textures was established as the control group. Based on the results of the finite element model, the results of the temperature field were numerically analyzed using the ANSYS general postprocessor and time history postprocessor. First, the basic characteristics of the joint temperature field were studied to clarify the thermal effect and forming mechanism of the joint in the welding process. Second, the temperature field distribution of each finite element model and the temperature change at a specific time and a specific path were compared to explore the influence of different scanning spacing micro-textures on the temperature field of the CFRTS-TC4 alloy laser bonding process.

## 3. Results and Discussions

### 3.1. Basic Characteristics and Analysis of Temperature Field

Figure 8 shows the temperature field distribution of the weld area on the surface of the TC4 alloy in a rotating cycle *T* of the heat source. It can be seen from the temperature cloud map that the moving trajectory of the rotating heat source is basically consistent with the actual welding trajectory, which indicates that the established heat source model is reliable. The temperature gradient of the weld zone is large, and the closer it is to the center of the heat source, the higher the temperature, and the denser the isotherm distribution. The maximum temperature at the center of the heat source is about 1881 °C.

To further study the temperature change law of the TC4 alloy surface during welding, five nodes were selected every 5 mm along the weld direction: *a*, *b*, *c*, *d*, *e* (Figure 9d). The relationship between the temperature of each node and time is shown in Figure 9a,b. As the rotating heat source moves, the temperature of each node on the surface of the TC4 alloy gradually increases or decreases in the form of broken line oscillation, and two temperature peaks appear on the same node. This is mainly due to the heat source being used for periodic circular motion; the welding process heat source irradiates the same node twice. When the heat source is close to the node, the node temperature gradually increases in the form of a broken line oscillation. When the heat source first reaches the node, the first temperature peak appears. When the heat source is far away from the node, the node temperature gradually decreases in the form of a broken line oscillation. When the heat source reaches the node for the second time, a second temperature peak appears. When the laser beam leaves the workpiece to stop heating, the temperature of each node gradually tends to be consistent. Since laser welding is a process of heat accumulation, the heat accumulated on the surface of TC4 alloy gradually increases over time, so the peak temperature from node *a* to node *e* gradually increases, and the second temperature peak of nodes *a*, *b*, *c,* and *d* is greater than the first temperature peak. Since node *e* is close to the edge of the workpiece, when the laser beam is far away from this point, the heat input is less than the heat loss, resulting in the second temperature peak being less than the first temperature peak.

To clarify the forming mechanism of the joint, five nodes, *f*, *g*, *h*, *i*, and *j*, were selected on the surface of CFRTS to observe the temperature change (Figure 9d). Figure 9c shows that the temperature change trend of each node on the surface of CFRTS is basically the same as that on the surface of the TC4 alloy, but there are no two obvious temperature peaks on the same node. This is due to the faster rotation speed of the laser; the heat is gradually homogenized when transferred inside the TC4 alloy, and the thermal conductivity of the PA6 resin is lower. When transferred to the CFRTS layer, more heat is lost. The numerical simulation results show that the maximum temperature of the selected nodes on the surface of CFRTS is higher than the melting point of PA6 resin during the whole welding process, indicating that PA6 resin can fully melt and flow into the gap of carbon fiber, and CFRTS can be fully combined with TC4 alloy to form welded joints.

### 3.2. Effect of Micro-Textures on Temperature Field of Joint

When the welding time *t* = 5 s is selected, the temperature field distribution on the weld of the TC4 alloy surface, PA6 interface, and CFRTS surface of each finite element model is compared. As shown in Figure 10, the temperature field distribution of the TC4 surface does not change due to the existence of micro-textures. The numerical simulation results show that the maximum temperature of the TC4 surface of each finite element model is about 1881 °C. At the PA6 interface, due to the presence of micro-textures, the isotherms on the interface are not smoothly distributed. This is mainly due to the fact that in the micro-texture area, the heat is concentrated on both sides of the micro-texture groove, resulting in an extremely uneven heat transfer to the PA6 interface. Without considering the finite element model of micro-textures, the heat diffuses evenly inside the material, and the distribution of the PA6 interface isotherm is smoother. The results of the numerical simulation show that the maximum temperature of the PA6 interface of each finite element model is about 418 °C. Due to the low thermal conductivity of PA6 resin, the distribution of the temperature field tends to be consistent when the heat continues to transfer downward to the surface of CFRTS.

Figure 11 shows a comparison of the temperature changes of the PA6 interface along the direction of the weld center line in each group of finite element models. It can be seen that the overall temperature change trend is basically the same, whereas the temperature change of the PA6 interface rises or falls in an oscillating form in the finite element model with micro-textures, and the temperature change of the PA6 interface is relatively stable in the finite element model without micro-textures. The difference in temperature change caused by micro-textures may affect the heat transfer from PA6 to CFRTS surface. Therefore, the temperature changes of each finite element model on the CFRTS surface along the center line of the weld were further compared to explore the effect of micro-textures on the heat transfer effect of the joint. Figure 12 shows that the temperature of the CFRTS surface is slightly higher than that of the finite element model without micro-textures, and the smaller the micro-texture size, the higher the temperature. This is because the groove of the micro-textures increases the contact area between the TC4 alloy and the PA6 resin, which increases the heat per unit area and the heat transferred downward, so the surface temperature of the CFRTS is higher.

### 3.3. Reliability Verification of Finite Element Model

To verify the reliability of the finite element model, a high-speed infrared thermal imager was used to capture the temperature distribution of the joint interface near the edge of the titanium alloy side at the end of welding, and the measured results were compared with the finite element simulation results. As shown in Figure 13, the actual measured temperature is slightly lower than that of the finite element simulation results. This is because, in the actual welding process, the thermal contact conditions between CFRTS and TC4 alloy are more complicated, and some random factors affect the experimental measurement results, resulting in errors. The overall temperature change trend is basically the same, and the error is within a controllable range. Therefore, the finite element model and numerical simulation results of CFRTS-TC4 laser welding are reliable.

### 3.4. Effect of Micro-Textures on Mechanical Properties of Joints

Laser micro-texturing on the surface of TC4 alloy is the key process for obtaining high-quality CFRTS-TC4 welded joints. In order to study the influence of different sizes of micro-textures on the properties of joints, high-speed rotating welding experiments of TC4 alloy and CFRTS laser based on laser micro-textures with different scanning spacings were carried out, and the tensile strength of the joint samples was tested using a universal material testing machine. Figure 14 shows the bonding strength of CFRTS and TC4 alloy joints with laser micro-textures at different scanning spacings. Among them, sample 1 was not laser micro-textured, and samples 2, 3, and 4 correspond to joints with micro-texture scanning spacing of 0.1 mm, 0.3 mm, and 0.5 mm, respectively. It can be seen that the strength of the joint without laser micro-texturing is significantly lower than that of the joint with laser micro-texturing. When the scanning spacing is 0.3 mm, the joint bonding strength is the highest, at about 14.3 MPa. Figure 15a shows the morphology of the joint interface when the micro-texture scanning spacing is 0.3 mm. It can be seen that the carbon fiber inside the CFRTS is in close contact with the plastic-covered layer of the TC4 alloy, and the CFRTS and the TC4 alloy form an interlocking structure at the joint interface, which is an important reason for the significant increase in joint strength. Because the laser micro-textures with a scanning spacing of 0.5 mm provide less interlocking structure, the connection strength of the joint is not significantly improved. In addition, when the scanning spacing is 0.1 mm, the removal of TC4 alloy material increases, and the micro-textures cannot be well formed (Figure 5a), which has a certain negative impact on the mechanical properties of the joint. The joint strength is lower than that of the laser micro-textured joint with a scanning spacing of 0.3 mm. The connection interface of the joint without laser micro-texturing does not form an obvious interlocking structure (Figure 15b), so the connection strength of the joint is the lowest.

To further clarify the fracture mechanism of the joint, the fracture surface morphology of sample 3 and sample 4 was observed using scanning electron microscopy. It can be seen from Figure 16a,b that when the scanning spacing of laser micro-textures is 0.3 mm, the carbon fiber on the surface of CFRTS is seriously broken after the fracture of the joint, and there is PA6 resin residue. A large amount of PA6 resin and carbon fiber are attached to the surface of the TC4 alloy. This indicates that the fracture interface of the joint is the bonding interface between CFRTS and TC4 alloy. The failure mode is mainly the cohesive failure of the internal tear of carbon fiber and the interface failure of the bonding interface between PA6 resin and TC4 alloy. According to Figure 16c,d, when the scanning spacing of laser micro-textures is 0.5 mm, a large amount of PA6 resin is attached to the surface of CFRTS after the fracture of the joint, and the carbon fiber is also damaged to varying degrees. However, only PA6 resin remains on the surface of the TC4 alloy, and no carbon fiber is found to be peeled off from the surface of CFRTS. This indicates that fracture of the joint mainly occurs on the interface between the carbon fiber layer and the PA6 resin, and the failure mode is mainly the interface failure of the carbon fiber and the PA resin tearing. Therefore, when the failure mechanism of the joint is the cohesive failure of the internal tearing of the carbon fiber and the mixed failure mode of the tearing of the interface between the PA6 resin and the TC4 alloy, it shows that the CFRTS is fully bonded with the TC4 alloy, and the joint strength is larger, which is basically consistent with the research results of Zou [27].

## 4. Conclusions

In this paper, a finite element model of high-speed rotating laser welding of CFRTS and TC4 alloy was established. The temperature field distribution of the joint during the welding process and the influence of laser micro-textures on the temperature field of the joint were studied. Through a high-speed rotary laser welding experiment of CFRTS and TC4 alloy, the influence of laser micro-textures with different scanning spacings on the mechanical properties of the joint was analyzed, and the reliability of the finite element model was verified. The main conclusions are as follows:(1)The finite element simulation results of the temperature field show that two temperature peaks appear at the nodes on the weld centerline of the TC4 alloy surface due to the periodic rotary heating of the TC4 alloy by the laser beam, and that the temperature of each node changes with time in the form of broken line oscillation. The temperature change trend of the nodes on the center line of the weld on the CFRTS surface is basically the same as that on the TC4 alloy surface. However, due to the fast rotation speed of the laser, the heat is gradually homogenized when it is transferred downward, and there are no two obvious temperature peaks at each node on the CFRTS surface.(2)Considering the finite element model of laser micro-textures, the isothermal line of the PA6 interface is not smoothly distributed, and the temperature along the weld center line changes in an oscillating form. Without considering the micro-textures of the finite element model, the PA6 interface isotherm distribution smoothed along the weld center line direction of the temperature change smoothly. Micro-textures cause a change in heat input on the surface of CFRTS, and the smaller the micro-textures, the higher the heat input on the surface of CFRTS. This is because the micro-texture groove increases the contact area between TC4 alloy and PA6 resin, which is conducive to heat conduction.(3)When the micro-texture scanning spacing is 0.3 mm, the bonding strength of the joint is the highest, about 14.3 MPa. The interlocking structure formed by CFRTS and titanium alloy at the interface is an important reason for the significant improvement in joint strength. At the highest bonding strength, the failure mode of the joint is the cohesive failure of carbon fiber tearing in CFRTS and the interfacial failure of PA6 resin and TC4 alloy.

## Figures and Tables

**Figure 1 materials-16-00270-f001:**
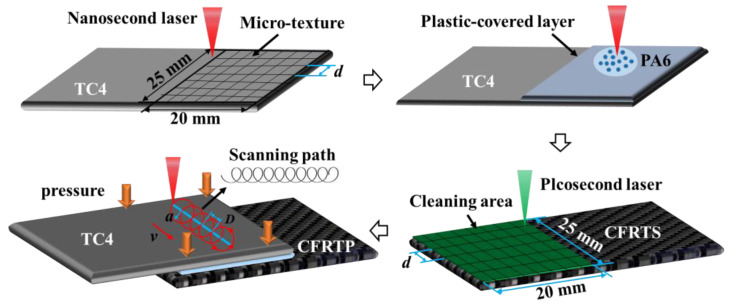
The interface hybrid pretreatment process [26,27].

**Figure 2 materials-16-00270-f002:**
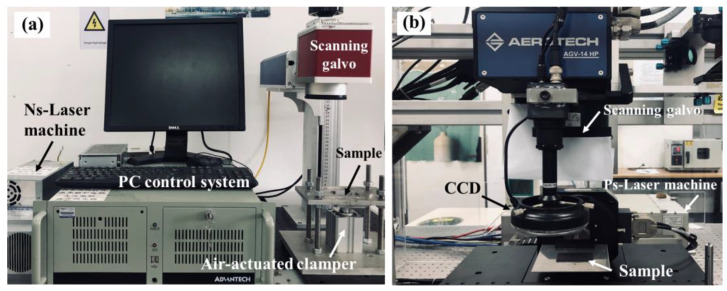
(**a**) Nanosecond laser machining system; (**b**) picosecond laser machining system [27].

**Figure 3 materials-16-00270-f003:**
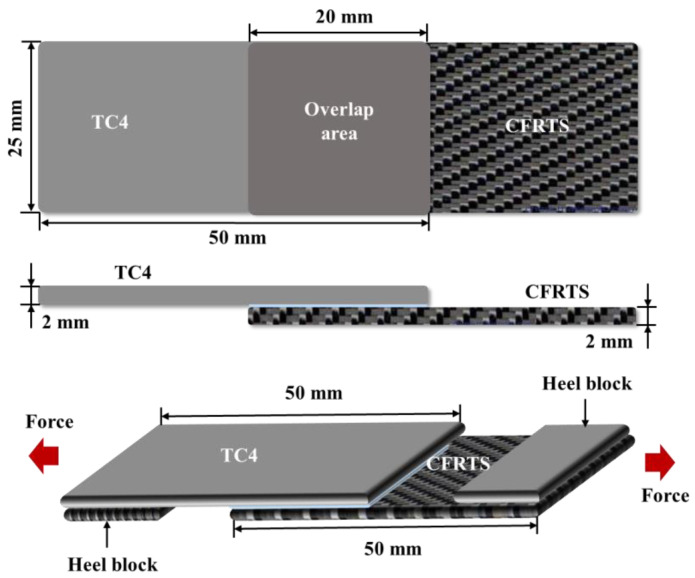
CFRTS-TC4 tensile sample.

**Figure 4 materials-16-00270-f004:**
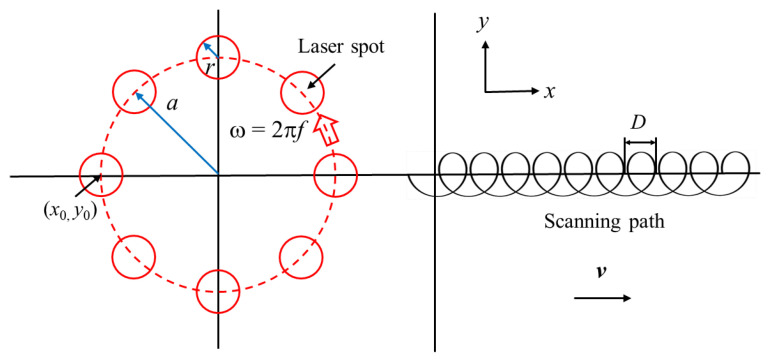
Heat source model and moving path.

**Figure 5 materials-16-00270-f005:**
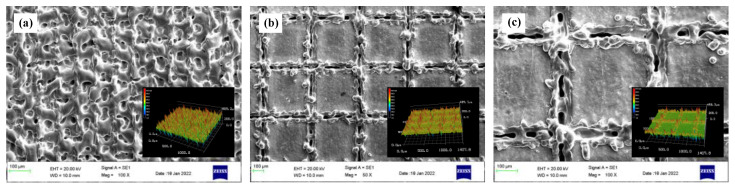
Surface morphology of TC4 alloy after laser micro-texturing: (**a**) *d* = 0.1 mm; (**b**) *d* = 0.3 mm; (**c**) *d* = 0.5 mm.

**Figure 6 materials-16-00270-f006:**
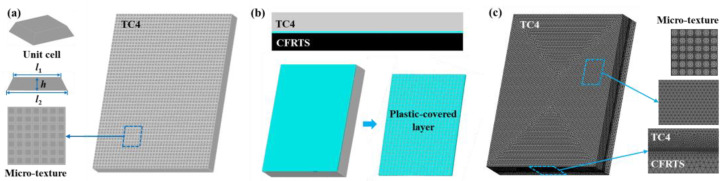
(**a**) Micro-texture geometric model; (**b**) anchorage model; (**c**) meshing model.

**Figure 7 materials-16-00270-f007:**
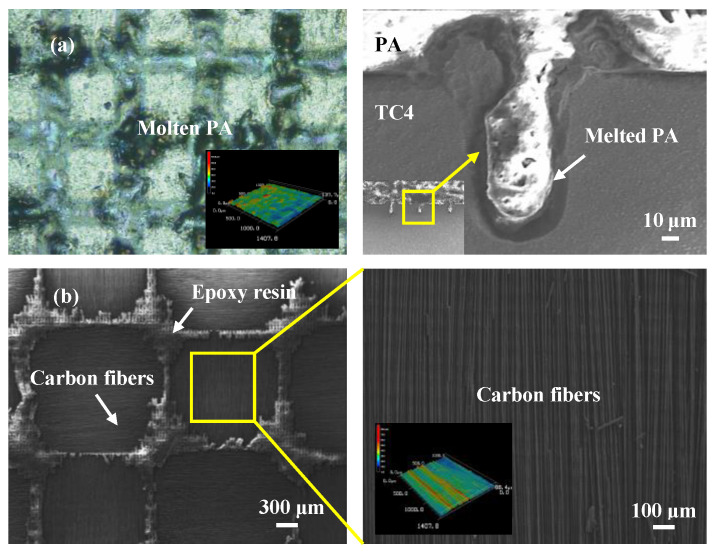
(**a**) Surface morphology of TC4 alloy after laser plastic-covered treatment; (**b**) surface morphology of CFRTS after laser cleaning.

**Figure 8 materials-16-00270-f008:**
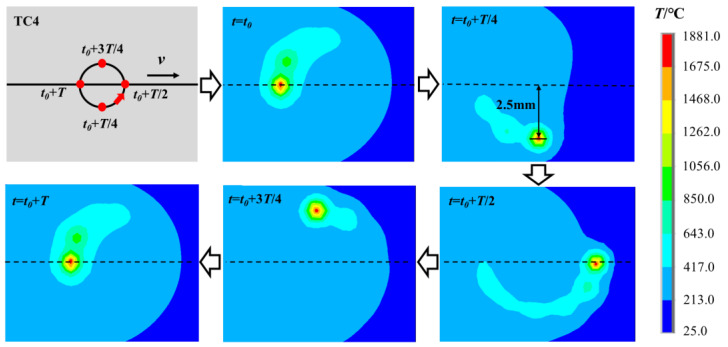
Temperature field of TC4 surface weld area in one cycle of rotating heat source.

**Figure 9 materials-16-00270-f009:**
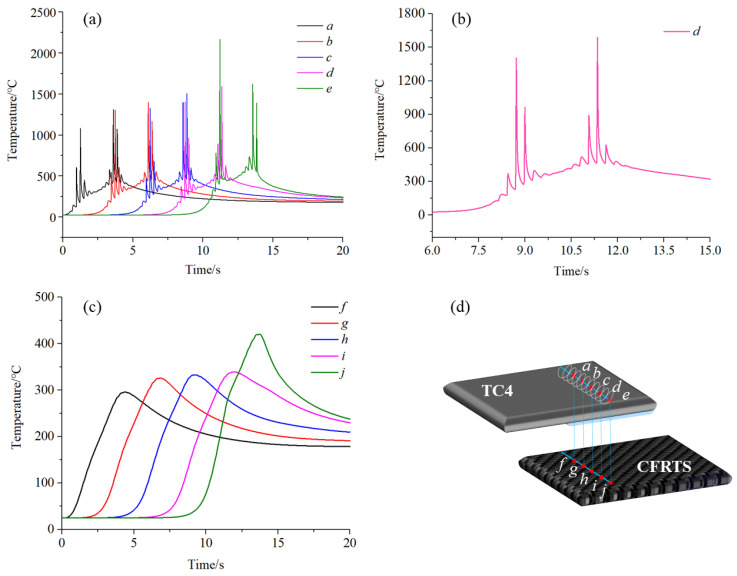
(**a**) Variation of temperature at each node of the TC4 surface over time; (**b**) the change of node *d* temperature with time; (**c**) variation of temperature at each node of the CFRTS surface over time; (**d**) node selection diagram.

**Figure 10 materials-16-00270-f010:**
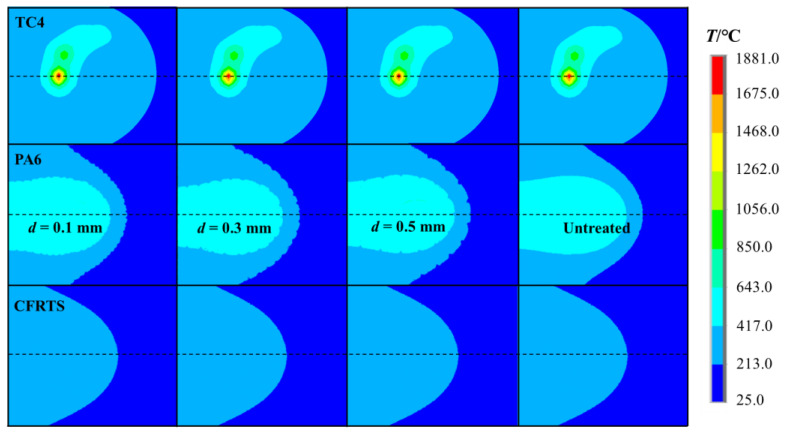
Comparison of the temperature fields of finite element models in each group.

**Figure 11 materials-16-00270-f011:**
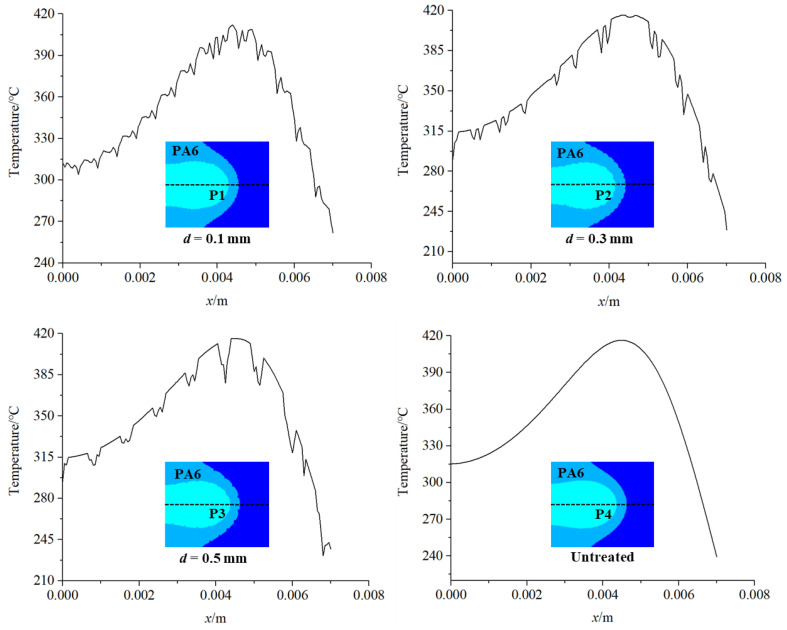
Comparison of PA6 interface temperature change.

**Figure 12 materials-16-00270-f012:**
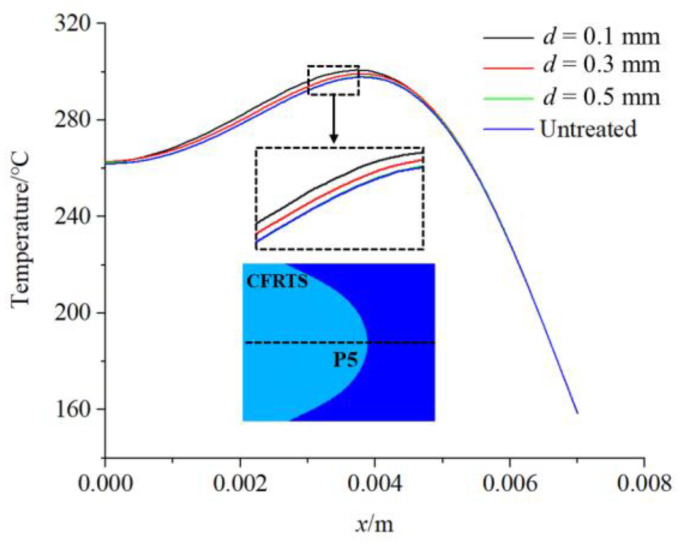
Comparison of CFRTS surface temperature change.

**Figure 13 materials-16-00270-f013:**
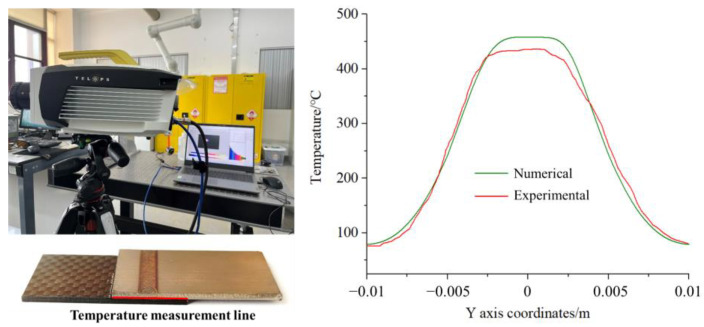
Comparison between experimental results and finite element simulation results.

**Figure 14 materials-16-00270-f014:**
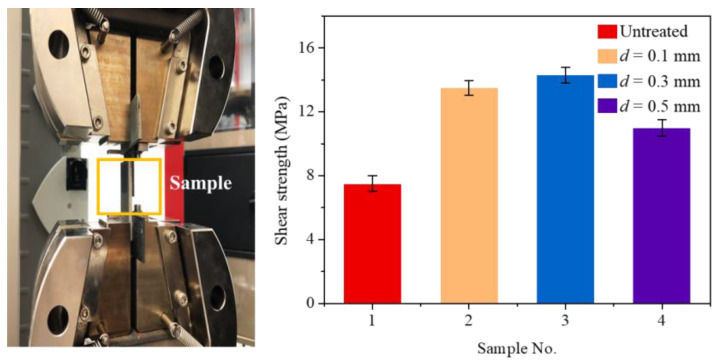
CFRTS-TC4 alloy joint tensile strength test.

**Figure 15 materials-16-00270-f015:**
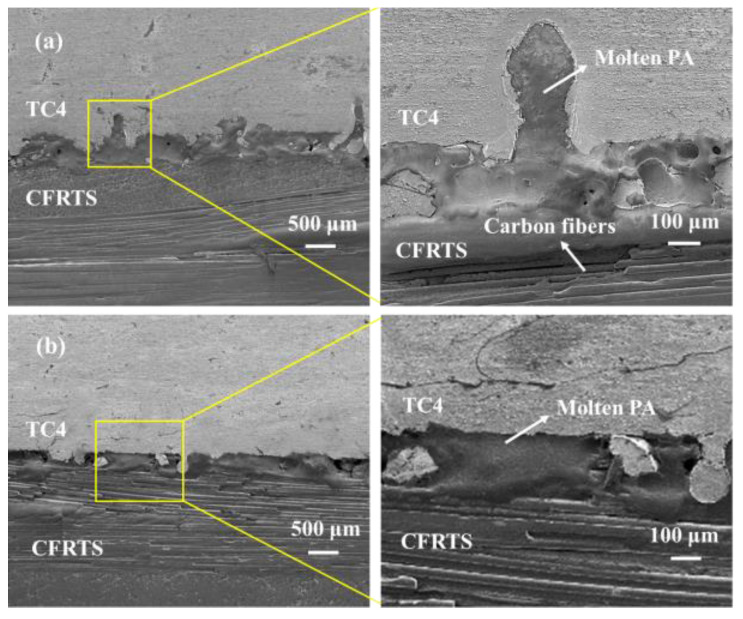
(**a**) Sample 3 interface morphology; (**b**) sample 1 interface morphology.

**Figure 16 materials-16-00270-f016:**
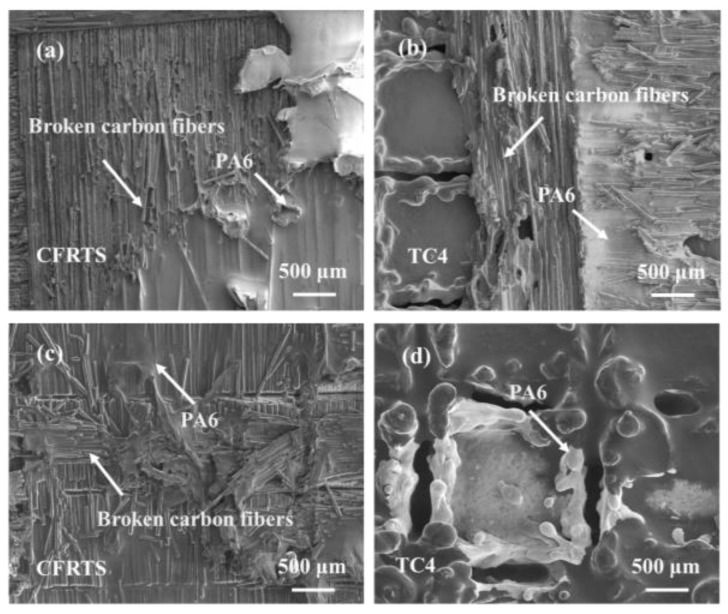
(**a**,**b**): Fracture surface morphology of sample 3; (**c**,**d**): fracture surface morphology of sample 4.

**Table 1 materials-16-00270-t001:** Thermophysical parameters of epoxy resin, T300, and PA6.

Material	Thermal Conductivity (W·m^−1^·°C^−1^)	Specific Heat (J·kg^−1^·°C^−1^)	Density (g·cm^−3^)
Epoxy resin	0.2	550	1200
T300	10.5	795.5	1760
PA6	0.25	2500	1150

**Table 2 materials-16-00270-t002:** Thermal physical parameters of TC4 alloy at different temperatures.

Temperature (°C)	Thermal Conductivity (W·m^−1^·°C^−1^)	Specific Heat (J·kg^−1^·°C^−1^)	Density (g·cm^−3^)
25	7.0	546	4420
200	8.75	584	4395
600	14.2	673	4336
1000	18.3	641	4282
1400	24.6	714	4225
1650	28.4	759	4189

**Table 3 materials-16-00270-t003:** CFRTS-TC4 laser high-speed rotational welding parameters.

*P* (W)	*V* (mm·s^−1^)	*D* (mm)	*a* (mm)	*r* (mm)	*P*_n_ (MPa)
95	50	0.5	2.5	0.15	0.8

**Table 4 materials-16-00270-t004:** Measurement dimensions of micro-textures at different scanning spacings.

*d* (mm)	*l*_1_ (mm)	*l*_2_ (mm)	*h* (mm)	*w* (mm)
0.1	0.072	0.116	0.095	0.047
0.3	0.211	0.298	0.096	0.049
0.5	0.423	0.497	0.093	0.045

## Data Availability

Data sharing not applicable.

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
