# Peer review of "Effect of Laser Micro-Texturing on Laser Joining of Carbon Fiber Reinforced Thermosetting Composites to TC4 Alloy"

_materials, 2022, doi:10.3390/ma16010270_

Round 1
Reviewer 1 Report
Name of the paper: Effect of laser micro-texturing on laser joining of carbon fiber reinforced thermosetting composites to TC4 alloy
This paper investigates the effect of laser micro-texturing on the laser joining of carbon fiber-reinforced thermosetting composites to TC4 alloy. Using a finite element model of CFRTS-TC4 alloy laser welding can predict the temperature field distribution of the joint during the welding process, and can reflect the forming mechanism of the joint. The laser scanning spacing is 0.3 mm, the joint strength can reach 14.3 MPa. The failure mechanism of the joint mainly includes the cohesive failure of the internal tear of the carbon fiber and the interfacial failure of the interface between the PA6 resin and the TC4 alloy.
General Observation:
1) Please refer to 422/423: “A large amount of PA6 resin and carbon fiber stripped from the surface of 422 CFRTS are attached to the surface of TC4 alloy.” The statement doesn't provide any clarity for the amount of PA6 resin stripping off. Stripped area or amount of PA6 resin stripped may help the further investigation for the breaking of carbon fiber.
2) Please refer to Figure 14 which shows the bonding strength of CFRTS and TC4 alloy joints with laser micro-textures at different scanning spacing. The graph shows the shear strength versus the sample number, it would have been better to have it against the scanning spacing for more clarity.
3) The study of Zou et al., 2022 observed ‘tearing off’ as a fracture mode at the CFRTS interlayer. The author may explain their results with possible failure mechanisms observed during the present experimentation.
4) Please refer to Figure 14, the shear strength observed without laser micro-texturing and laser micro-texturing with 0.5 mm thickness is 3MPa only, authors may explain these results.
5) Please refer to line number 184: “Based on the previous experimental research ….he laser power, scanning speed, scanning spacing and scanning times of laser plastic-covered are selected as 35 W, 200 mm·s-1, 0.3 mm and 5 respectively.” (In previous research P was considered as 25W,35W, and 45W, Table 4 Experimental design for TC4 surface plastic-covered pretreating.) The author may provide some more personal experience with the results obtained in its selection to help future researchers.
6) The comparison of joint strength with solid TC4 or CFRTS would have been adding more clarity.
7) The simulation model has been validated for either TC4 or CFRTS temperature zone discreetly. The simulation model validation for joints is unclear.
8) It is seems from the joint morphology that titanium alloy had cracks and the resin deposited into the cracks. Since the joint is already treated before joining, there are minimal chances for cracks. The author may provide some explanation to justify the presence of resin in the TC4 portion
Reviewer 2 Report
1- More details, including the model, must be added regarding the laser systems that are being used.
2- The originality of this work and how it differs from earlier research should be explained by the authors.
3- What effects might laser polarisation and wavelength have on the laser bonding of CFRTS and TC4 alloy?
4- Does the finite element model include the laser wavelength and laser polarisation parameters?
5- Nothing is mentioned in the text about the uncertainty in the measurements, so please discuss this issue in the text.
6- In addition to the References list, there are more recent work is suggested to be included as
· "Novel Surface Topography and Microhardness Characterization of Laser Clad Layer on TC4 Titanium Alloy Using Laser-Induced Breakdown Spectroscopy and Machine Learning" Metallurgical and Materials Transactions A volume 53, pages3639–3653 (2022).
· “A. Laser Micro-Texturing of Sintered Tool Materials Surface.” Materials 2019, 12, 3152. https://doi.org/10.3390/ma12193152
Round 2
Reviewer 2 Report
The authors have made satisfactory amendments to the manuscript in response to my previous comments and concerns. Overall the manuscript reads well and has clarified the work of the authors. In my opinion, the manuscript contains now all information and is suitable for publication in the Journal “materials” as a regular article.